# Design of Green Silver Nanoparticles Based on *Primula Officinalis* Extract for Textile Preservation

**DOI:** 10.3390/ma15217695

**Published:** 2022-11-01

**Authors:** Mihaela Cristina Lite, Rodica Roxana Constantinescu, Elena Cornelia Tănăsescu, Andrei Kuncser, Cosmin Romanițan, Ioana Lăcătuşu, Nicoleta Badea

**Affiliations:** 1Faculty of Chemical Engineering and Biotechnology, University Politehnica of Bucharest, 1-7, Polizu Street, 011061 Bucharest, Romania; 2National Research and Development Institute for Textiles and Leather–INCDTP, Lucretiu Patrascanu 16, 030508 Bucharest, Romania; 3National Institute of Materials Physics, Atomistilor 405A, 077125 Magurele, Romania; 4National Institute for Research and Development in Microtechnologies, Erou Iancu Nicolae 126A, 077190 Voluntari, Romania

**Keywords:** *Primula officinalis*, green synthesis, silver nanoparticles, textile preservation

## Abstract

The present study aims to bring an addition to biomass resources valorization for environmental-friendly synthesis of nanoparticles. Thus, the green synthesis of silver nanoparticles (AgNPs) was performed, using a novel and effective reducing agent, Primula officinalis extract. The synthesis was optimized by monitoring the characteristic absorption bands, using UV–Vis spectroscopy, and by evaluating the size and physical stability. The phenolic consumption was established using Folin-Ciocâlteu method (1.40 ± 0.42 mg, representing ~5% from the total amount of poly--phenols) and the antioxidant activity was evaluated using chemiluminescence and TEAC methods. The optimum ratio extract to Ag ions was 1:3, for which the AgNPs presented a zeta potential value of −29.3 ± 1.2 mV and particles size of 5–30 nm. For characterization, EDS and XRD techniques were used, along with microscopy techniques (TEM). The AgNPs dispersions were applied on natural textile samples (cotton and wool), as a novel antimicrobial treatment for textile preservation. The treated fabrics were further characterized in terms of chromatic parameters and antimicrobial effect against Escherichia coli, Staphylococcus aureus, Bacillus subtilis, and Penicillium hirsutum strains. The high percentages of bacterial reduction, >99%, revealed that the AgNPs produced are a good candidate for textiles preservation against microbial degradation.

## 1. Introduction

The field of nanomaterials has taken a huge step forward with the advent of green synthesis methods [1]. The research in this area has contributed significantly to overco-ming the disadvantages of the conventional methods, which consist of chemical, physical and photochemical approaches that require a high energy consumption and generation of hazardous waste [2,3]. The green alternatives promote a series of considerable advantages in terms of environmental friendliness, through biomass valorization, limiting the use of chemical reagents, hence decreasing the costs and toxicity, without compromi-sing the efficiency [4]. Literature abounds in research studies reporting different components of biomass resources involved in nanoparticles synthesis. The strategy of this method consists in replacing the traditional reducing agents, such as NaBH_4_, with those naturally contained in biological structures, such as enzymes, bacteria, fungi, yeast, algae, and plant extracts, as they exhibit an excellent reducing potential due to their abundant phenolic content [5,6]. Among the nanomaterials studied by the scientific community, silver nanoparticles have gained a particular attention, due to their applicability in a broad spectrum of research areas, e.g., catalysis [7,8,9], sensors [10,11], optoelectronics [12], biomedical applications [13] etc. Moreover, AgNPs earned a strong reputation as disinfec-tants due to their remarkable antimicrobial properties, with application in medicine (wound healing, drug delivery, medical devices etc.) [14,15], as much as in environmental protection (e.g., water disinfectant) [16] and other industries (e.g., food packaging and products disinfection) [17]. Hence, different approaches to the synthesis of silver nanoparticles in a green manner have been intensively studied [18]. Furthermore, the efforts for improving their bioperformances are under continuous investigation, e.g., by developing green AgNPs based nano coatings or hybrid structures, using pectin or biomimetic membranes, for enhancement of antioxidant, antibacterial, and hemolytic activity enhancement [19,20].

Textile objects, especially those manufactured from natural fibers, are susceptible to microbial degradation, leading to depolymerization, loss in breaking resistance and fading [21]. In the process of textiles biodegradation, populations of microorganisms growing on the surface of the fibers release metabolic products, such as enzymes and free radicals, breaking the macromolecules into oligomers, and monomers [22]. Thus, the concept of textiles functionalized with antimicrobial silver nanoparticles-based agents came as a potential solution for their preservation [23]. Several studies reported the in situ production of AgNPs directly on textile fabrics, by immersing the samples in the AgNO_3_ solution before adding the reducing agent. When using conventional reducing agents, the main disadvantage is related to the dark color that the textile samples acquire after the procedure [24,25,26,27]. This aspect was partially overcome by using green reducing agents, such as *Seidlitzia rosmarinus*, when the samples turned yellow instead of grey [28]. Nevertheless, the in situ synthesis on textile fabrics required an alkali pre-treatment of the fabrics, to activate them prior to AgNPs attachment. Further studies showed that by simply applying AgNPs colloidal suspensions on textile samples, via immersing or spraying, with or without a binder, the antimicrobial activity is maintained, even after washing and without performing a pre-treatment [29]. Also, Gutarowska and collaborators proposed an effective method of treating textile artefacts against microbial degradation, using a misting chamber in which silver colloid is sprayed from four sides [30,31,32].

In this paper, the synthesis of AgNPs through a green method is reported, using the *Primula officinalis* plant extract, also known as primrose. Primrose is a medicinal herb highly widespread throughout Europe and Asia, with a generous content of flavonoids, useful for promoting a successful reducing reaction [33,34]. According to Tarapatskyy et al. the main polyphenolic compounds from *Primula veris*/*Primula officinalis* are consisting in 18 Quercetin- and Kaempferol-glycosidic derivatives [34]: 5-O-(E)-caffeoyl-galactaric acid; Dicaffeoyl-protocatechuic acid diglucoside; Quercetin 3, 7, 4′-O-triglucoside; Quercetin-3-O-diglucoside; Quercetin 3-O-rutinoside-7-O-rhamnoside; Quercetin 3-O-rutinoside-7-O-glucoside; Quercetin 3-O-diglucoside-7-O-glucuronide; Kaempferol 3-O-rutinoside-7-O-rhamnoside; Quercetin 3-O-glucoside-7-O-rhamnoside; Quercetin 3-O-rutinoside (Rutin); Quercetin 3-O-glucoside; Quercetin 4′-O-glucoside; 6,3′-dimetoxyquercetin 7-O-diglucoside; Kaempferol 3-O-glucoside-7-O-rhamnoside; Kaempferol 3-O-rutinoside; Quercetin 3-O-glucuronide-7-O-rhamnoside; Isorhamnetin 3-O-rutinoside; Quercetin 3-O-glucuronide. In the literature, the synthesis of silver nanoparticles involving the use of phytochemicals as reducing agents has been reported in an extensive manner [35,36,37,38]. However, no research has addressed the involvement of *Primula officinalis* plant extract as an effective reducing agent for obtaining metallic nanoparticles.

In order to obtain optimized AgNPs, the green synthesis was performed in different ratios of extract to silver ions and the production of AgNPs was monitored by observing the formation of the specific SPR band (Surface Plasmon Resonance) in the UV–Vis spectra [39]. The resulting dispersions were characterized in terms of size and polydispersity, as much as their physical stability, by following the behavior of the surface potential (zeta potential). The antioxidant activity of the dispersions enriched in silver nanoparticles was assessed, to evaluate how it could influence the scavenging of two kinds of free radicals of short and long-life (oxygenated radicals and ABTS cationic radicals). For the first time, in the present study, the total amount of polyphenols consumption, necessary for the effective reduction of silver anions to metallic silver, was calculated. The synthesized AgNPs dispersion was applied to textile fabrics (cotton and wool). This type of preservation treatment for textiles, based on AgNPs produced using *Primula officinalis* plant extract, represents the novelty of this work. The antimicrobial effect of the applied AgNPs dispersion was tested against different bacteria and fungi strains, and the treated textile samples were further characterized by microscopic techniques and in terms of chromatic parameters.

## 2. Materials and Methods

### 2.1. Materials

Dried primrose plant was purchased from the producer StefMar (Râmnicu Vâlcea, Romania) and silver nitrate from Anal-R NORMAPUR (Radnor, PA, USA). Trolox (6-hydroxy-2,5,7,8-tetramethylchroman-2-carboxylic acid), 2,2azinobis–(3-ethylbenzthiazoline-6-sulfonic acid) (ABTS), gallic acid, anhydrous sodium carbonate, potassium persulfate and NaCl were purchased from Sigma Aldrich (Saint-Louis, MO, USA) and Tris (hydroxymethylaminomethane base), HCl, H_2_O_2_, luminol (5-amino-2, 3-dihydro-phthalazine-1, 4-dione), from Merck (Darmstadt, Germany).

Textile samples were provided by the National Research and Development Institute for Textiles and Leather–INCDTP (Bucharest, Romania). The antimicrobial activity tests were conducted using TSB (Tryptic Soy Broth), TSA (Casein Soya Bean Digest Agar), EA (Enumeration Agar), NB (Nutrient broth) and SCDLP (Casein Soya Bean Digest) culture media) and *Staphylococcus aureus* ATCC 6538), *Escherichia coli* ATCC 10536), *Bacillus subtilis* ATC 6633), and *Penicillium hirsutum* ATCC 52323 strains).

### 2.2. Synthesis Optimization of Silver Nanoparticles

Primrose extract was prepared in two concentrations, by infusing one and three grams, respectively, of dried plant in 100 mL boiled water. After 30 min, the mixtures were cooled to room temperature and filtrated. For the AgNPs synthesis, aqueous solution of AgNO_3_ (1 mM) has been added to the prepared extract, in different ratios. All dispersions were prepared in transparent containers, in a sunlight simulating chamber, as the reaction is photosensitive. Afterwards, the dispersions were stored in brown containers. To study how the reaction is influenced by the light exposure, the synthesis was performed in parallel both in the simulated sunlight and in the darkness and the UV-Vis spectra were recorded at different times of the reaction.

### 2.3. Physico-Chemical and Biological Characterization of Silver Nanoparticles 

#### 2.3.1. Spectral Characterization of Silver Nanoparticles

UV–Vis absorption spectroscopy was performed using a Lambda 950 instrument from PerkinElmer) (Waltham, MA, USA), on a spectral range of 200–700 nm. The optical properties of silver colloidal suspensions arise from their nanometric dimensions as the electron density oscillates on the particle surface, absorbing electromagnetic radiation. This oscillation, called surface plasmon resonance (SPR), is an excellent indicator of the formation of AgNPs [39].

#### 2.3.2. Physical Stability, Size, and Morphology Characterization of Silver Nanoparticles

The average size of the silver nanoparticles formed was evaluated by dynamic light scattering (DLS) using a Zetasizer NanoZS (Malvern Instruments Inc., Worcestershire, UK). The measurements were performed at 25 °C, in three sets for each sample and the mean values were reported. Sample preparation consisted of pipetting 300 µL of AgNPs dispersion to 20 mL distilled water, and the resulting suspension was analyzed by DLS technique. Furthermore, the distribution of the particles populations as a function of size is given by the polydispersity index. 

Zeta potential (ξ) is an indicator of the physical stability, and for a dispersion to be considered stable, the absolute value of its zeta potential should exceed 25 mV [40]. To evaluate the stability of the AgNPs dispersions an electric field was applied across the tested suspensions, to which 50 μL of NaCl 0.9% solution was added, and the zeta potential was measured using the Malvern Zetasizer Nano ZS (Malvern Instruments Inc., Worcestershire, UK) equipment, as previously presented. Each suspension was triple measured, and the mean values were reported.

The morphological characterization was conducted via high-resolution transmission electron microscopy (TEM), using the instrument Cs probe-corrected JEM ARM 200F) (JEOL Ltd., Tokyo, Japan) analytical electron microscope, and scanning electron microscopy (SEM)), using a FEI Quanta 200 microscope) (ThermoFisher Scientific, Waltham, MA, USA), equipped with an Everhart-Thornley (ET) detector), at an accelera-ting voltage of 15 kV, in low vacuum mode. Moreover, their nature was confirmed through X-ray dispersive spectroscopy (EDX), by coupling an X-ray detector from EDAX-AMETEK) (Berwyn, PA, USA) to the electronic microscope. X-ray diffraction (XRD) was performed using a 9 kW Rigaku SmartLab instrument) (Rigaku, Tokyo, Japan), equipped with a Cu Kα1 source (λ = 0.154 nm). XRD patterns were recorded in a grazing incidence mode, for which the incidence angle, ω was fixed at 0.5°, while 2θ varied from 35 to 80°.

#### 2.3.3. Phenolic Content Assessment

The phenolic content of both extract solutions and AgNPs dispersions was determined by the colorimetric method for total polyphenols content determination using Folin-Ciocâlteu reagent) (Merck, Darmstadt, Germany), according to ISO 14502-1:2005 [41]. The calibration curve was constructed using gallic acid standard solutions, in the concentration rage from 0 to 60 µg/mL (R² = 0.9997). For the analysis, 0.5 mL of each sample (extract solution and AgNPs dispersion, respectively) and 4.5 mL of anhydrous Na_2_CO_3_ solution 7.5% (m/m) were added to 5 mL Folin-Ciocâlteu reagent 10% (*v*/*v*). The mixtures were incubated in the darkness at room temperature for 1 h, then their absorbance was measured spectrophotometrically, at λ = 765 nm. All samples were measured in triplicate, and the phenolic content was expressed as gallic acid (GA) equivalent. Furthermore, by subtracting the phenolic content of the extract solutions and AgNPs dispersion, it is possible to calculate the polyphenol’s average consumption.

#### 2.3.4. In Vitro Evaluation of Antioxidant Activity

The antioxidant behavior of the AgNPs dispersions was evaluated and compared to the diluted extract solutions by chemiluminescence and TEAC (Trolox Equivalent Antioxidant Capacity) assays. The extract solutions were prepared by diluting the extract used for AgNPs synthesis, with respect to the ratios extract: AgNO_3_ in which they were combined.

The chemiluminescence (CL) measurements were performed, in triplicate, on a Turner Design TD 20/20 USA Chemiluminometer (Sunnyvale, California, USA). The generator system of the short-life oxygen free radicals (ROS) consisted of luminol (0.01 mM) in TRIS-HCl buffer solution (pH 8.6), and H_2_O_2_ (0.01 mM). This reaction mixture was used as a reference [42]. The antioxidant activity (*%AA*) was calculated as follows:(1)$$\%AA=\frac{I_{0}-{\;I}_{s}}{I_{s}}\cdot 100$$
where I_0_ and is the maximum CL intensity for the blank sample and I_s_ the maximum CL intensity for the tested samples.

The antioxidant activity evaluated through TEAC assay involves the spectral monitoring of the long-life radical ABTS^●+^, which is formed by reaction between ABTS solution (7 mM) and potassium persulfate solution (2.45 mM). The resulting solution is norma-lized, after 16 h, at 734 nm to an absorbance value of 0.70 (± 0.02). The spectral measurements were performed using an UV-Vis-NIR Spectrophotometer (V670, Jasco, Tokyo, Japan). A calibration curve was constructed using standard Trolox solutions prepared in the concentration rage from 0 to 60 µM (R^2^ = 0.9990). The inhibition percentage of ABTS^●+^ cation radical was calculated as follows:(2)$${\%\mathrm{Inh}\;\mathrm{ABTS}}^{\cdot +}=\frac{A_{0}-{\;A}_{s}}{A_{0}}\cdot 100$$
where A_0_ is the absorbance of the blank (3 mL of ABTS^●+^ diluted solution and 2 mL of distilled water) and A_s_ is the absorbance of the samples (3 mL ABTS^●+^ diluted solution, 0.5 mL AgNPs dispersion/diluted extract + 1.5 mL of distilled water). All samples were measured in triplicate, and the antioxidant capacity was expressed as Trolox equivalent.

### 2.4. Characterization of Textile Samples Treatment Based on AgNPs Dispersion

Cotton and wool samples were treated with the AgNPs dispersion prepared in the optime ratio. The treatment method consisted of immersing 10 cm × 10 cm pieces of textile fabrics in the AgNPs dispersions and letting them to dry overnight.

#### 2.4.1. Evaluation of Chromatic Parameters

The chromatic were measured using a Datacolor instrument (Datacolor, Inc., Lucerne, Switzerland). The chromatic parameters were expressed in the CIE L*a*b* system of colors, where L* parameter refers to the luminosity of the samples, and parameters a* and b* describe the color of the samples. L* has values in the range 0–100, where 0 is for black and 100 is for white. Parameters a* and b* have values between −100 and +100 and, in an xOy axis system (where a* represents the Ox axis and b* the Oy axis), they indicate the color as follows: if a* and b* are positive, the color of the sample will be in the range of red-orange-yellow. If a* is negative and b* is positive, the color of the sample will be in the yellow-greenish-green range. If a* and b* are negative, the color of the sample will be in the range green-turquoise-blue. If a* positive and b* negative, the color of the sample will be in the range of blue-purple-red. Having all the parameters for a control sample (untreated textile fabric) and for the treated samples, a total color changed (ΔE*) can be calculated, according to the formula: ΔE* = [(ΔL*)^2^ + (Δa*)^2^ + (Δb*)^2^]^1/2^ [43]. It is important to know the chromatic effect of the treatment used in the conservation-restoration of he-ritage objects, as it is necessary that it does not change the color of the artifacts or, in the case of restoration, slightly intensifies the color tones, for objects that have faded over time [44].

#### 2.4.2. Antimicrobial Activity of Textile Samples

The antimicrobial assays were performed in terms of antibacterial activity (against *Escherichia coli, Staphylococcus aureus*, and *Bacillus subtilis* strains) and antifungal activity (against *Penicillium hirsutum*).

In order to determine the antibacterial activity, the samples were tested using the ISO 20743: 2013 standard, which involves quantitative testing methods to determine the antimicrobial activity of the finished antimicrobial samples [45]. The absorption method was used, which involved direct inoculation of the test bacteriological inoculum directly onto the treated samples. The results were expressed as an average percentage. Counts on the plate were performed at 24 h of incubation, in order to determine the colony-forming units (CFUs).

The bactericidal ratio, *R* (%) was calculated as follows:(3)$$R\;\left(\%\right)=\frac{CFU_{control}-CFU_{sample}}{CFU_{control}}\times 100$$

A qualitative assessment was also performed for both bacteria and fungi strains, by agar well diffusion method [46,47]. The entire Petri dishes surface was inoculated by spreading a volume of each strain and the textile samples (10 mm diameter) were placed on the surface of the nutrient medium and then incubated at 37 °C for 24 h. The formation of a clear inhibition zone (*IZ*) indicates the antimicrobial effectiveness of the samples.

The inhibition zone is calculated according to the following formula:(4)$$IZ=\frac{D-d}{2}$$
where *D* is the total diameter of the sample and the inhibition zone (mm), and *d* is the diameter of the sample (mm).

### 2.5. Statistical Methods

All data were expressed as the mean value ± standard deviation of three individual experiments. Statistical significance was estimated using the student’s t–test (Microsoft Excel 2010) to determine the significant differences among the experimental groups, and values of *p* < less than 0.05 were considered statistically significant.

## 3. Results and Discussion

### 3.1. Evaluation of the AgNPs Dispersions

#### 3.1.1. Spectral Characterization of the AgNPs Dispersions

The silver nanoparticles synthesis was optimized starting from different extract concentrations (1 and 3 g/100 mL), varying the volume ratios between AgNO_3_ and primrose extract, the absence, or the presence of light. The first experiments involved a variation between primrose extract (3 g/100 mL): AgNO_3_ volume ratio, e.g., 9:1, 5:1, 3:1, 1:1, 1:3, 1:5, and 1:9 (*v*/*v*). After 24 h, the dispersions subjected to UV-Vis spectroscopy revealed the AgNPs formation, by the presence of the SPR (Surface Plasmon Resonance) band at 420–450 nm. It has been noticed in the overlapped absorption spectra (Figure 1a) that AgNPs formation is favored by increasing the silver ions concentration in the reaction mixture. As a result, additional experiments were reported by using more diluted concentrations of the extract, e.g., 1 g/100mL, in the following ratios 1:1, 1:2, 1:3, 1:4, 1:5, 1:9, 1:11, and 1:15 (extract: AgNO_3_). The UV-Vis overlapped spectra absorption is depicted in Figure 1b. The maximum absorbance of the SPR band was registered for the sample with 1:3 ratio (extract: AgNO_3_).

The influence of the light exposure on the reaction rate (for dispersion with the ration 1:3) at different times of the synthesis, both in the simulated sunlight and in the darkness, is depicted in Figure 2. The overlapped spectra revealed that the reaction is occurring very slowly in the absence of light, suggesting that the reaction is photocatalyzed. Moreover, by plotting the absorbance of the SPR bands as a function of time, the reaction rates were established, for the first 3 h (0.1318 h^−1^), between 3 and 8 h (0.0641 h^−1^), and between 8 and 32 h (0.0047 h^−1^), pointing to zero afterwards. 

The active components contained in the primrose extract acted both, as reducing and capping agents, and their molecular structure could drastically influence the shape and size of the silver nanoparticles. When higher concentrations of extract were used in the synthesis, the SPR band did not form in the UV–Vis spectrum (Figure 1a), even though the reaction mixture turned into a brownish color. This behavior suggests a strong reduction of the silver ions, followed by a fast aggregation, with the formation of large particles of colloidal silver, as indicated in previous reports [48,49]. The increase in the silver ions concentration (Figure 1b) led to the formation of the SPR band, at 434 nm, reaching the maximum intensity at 1:3 extract to Ag^+^ ions ratio (*v*/*v*). By further increasing the silver ions concentration the intensity of the SPR band decreases. This phenomenon was observed in other studies [50,51,52] and Velammal et al. suggested that it occurs due to the formation of Ag_2_O [49].

Light radiation is a key factor in the AgNPs synthesis. The intensity or the wavelength of the radiation can influence the size and shape of the nanoparticles, or reaction rates [53]. The publications in the field of green synthesized AgNPs demonstrate that the best results are obtained when the reaction is performed at sunlight exposure [54,55] and the present study sustains these results. Also, the increase in the SPR band intensity without modifications of its width indicates that the reaction is stable up to 72 h. For compa-rison, Hashemi et al. reported a green synthesis of AgNPs using *Sambucus ebulus* phenolic extract, where the sharp SPR band in the UV–Vis spectrum decreased after 4 h, suggesting that the reaction tended towards aggregation of AgNPs [56]. Also, when plotting the intensity of the absorbance versus time, it was found that the maximum reaction rate was maintained for the first 3 h. Consequently, this interval was selected as the optimal reaction time.

#### 3.1.2. Size and Polydispersity of the AgNPs

The average size values of the silver nanoparticles and the polydispersity indices (PDI) are illustrated in Figure 3. The average diameter of AgNPs was in the range 34–126 nm. The sample with ratio 1:1 presented the maximum particle size (126 nm), having a polydisperse index of 0.225. The minimum particle size (34 ± 0.888 nm) was recorded for the sample with 1:3 ratio, having a polydisperse index of 0.346.

Except for the first ratio (1:1), in all the other cases, the average size values are under 100 nm, situated in the interval 34–55 nm. Also, the PDI index is maintained around the value 0.34, except for the ratios 1:1, 1:2, and 1:15, at which it has values in the range 0.22–0.25. These results are comparable with those reported in literature for the synthesis of AgNPs using *Acorous calamus* rhizome extract–31.8 nm [57], turnip leaves extract (*Brassica rapa L.*)–39.5 nm [58], leaves of *Thelypteris glandulosolanosa*–48.1 nm (PDI–0.472) [59], *Melia azedarach*–78 nm [60], *Rubus rosifolius* fruit extract–85.6 nm (PDI–0.266) [61], garlic clove extract–90 nm [62], fresh leaves of *Mentha piperita*–88 nm (PDI–0.132), and *Amaranthus retroflexus*–92 nm (PDI–0.232) [20].

The nature of the particles was established by energy dispersive X-ray spectroscopy. The EDX spectrum is represented in Figure 4.

The peak corresponding to Ag appears typically at 3 keV [63]. Other elements, carbon, chlorine, and potassium were also present in the EDX spectrum, most probably ori-ginating from the phytogenic content of the plant extract [64].

TEM images on the sample synthesized with *Primula officinalis* extract 1:3 (Figure 5a,b) revealed a system of cvasi-spherical NPs with size frequently in the range 5–30 nm (Figure 5c). A small fraction of the AgNPs tend to agglomerate into large clusters (300 nm) with flower-like aspect. These irregularly shaped AgNPs from the clusters could be a consequence of particles agglomeration. A similar agglomeration phenomenon was previously reported by Hebeish and co–workers (2011) in their study regarding AgNPs synthesis with cyclodextrins, when using the thermal heating approach [65]. Also, Sánchez et al. (2016) reported the same behavior of Ag nanoparticles agglomeration [66].

#### 3.1.3. X-Ray Diffraction Analysis (XRD)

The diffractogram obtained (Figure 6) includes the diffraction lines of different planes of AgNPs: (111) at 2θ = 38.03°, (200) at 2θ = 44.35°, (220) at 2θ = 64.47°, and (311) at 2θ = 76.66°, with a crystalline packing of cubic silver (a = 0.41 nm, 225: Fm3m spatial group). The size of the crystalline domains, or mean crystallite size is: 6.4 nm (Scherrer equation)–based on (111) FWHM, 4.8 nm (Rietveld)–fitting of the whole pattern, and 4.4 nm (size-strain Williamson-Hall plot)–FWHM and peak positions corresponding to silver, and the small lattice strain was around +0.3%. The phase identification was achieved using the International Centre for Diffraction Data (ICDD) database (DB card no. 00-004-0783). The other diffraction lines present in the diffractogram correspond to NaCl, which can originate from the washing water.

Similar patterns were obtained when using *Melia azedarach* or *Eriobotrya japonica* (Thunb.) leaf extracts [67,68].

#### 3.1.4. Physical Stability of the AgNPs Dispersions

The stability of silver nanoparticles is directly proportional to the presence of negative charges on their surfaces, which is confirmed by the value of the zeta potential (ξ). The zeta potential values of the dispersions (Figure 7a) have ranged between −26.2 and −33.8 mV, proving that all samples are homogenous and physically stable. These values are within the range of those reported in literature, between −24.9 mV and −32.3 mV for AgNPs synthesized using turnip leaf extract (*Brassica rapa L.*) [58], *Melia azedarach* extract [60], *Rubus rosifolius* fruit extract [61], *Tridax procumbens* plant extract [69], fresh leaves of *Mentha piperita* and *Amaranthus retroflexus* [20]. The high stability of the nanoparticles is attributed to the phytogenic content, such as terpenes and polysaccharides, which can act as surfactants [70,71]. The distribution of the zeta potential for ration 1:3 is depicted in Figure 7b.

#### 3.1.5. Evaluation of Phenolic Consumption

The reduction of silver ions occurs according to Figure 8, in the presence of polyphenols contained in the extract solutions. By determining the total polyphenols in extract solutions and in the AgNPs dispersions (Figure 9a), a linear dependency of the phenolic content on the extract concentration was observed (Figure 9b), suggesting that, at all ratios, the same amount of phenolic content is consumed in the reaction.

The total amount of phenolic content in the extract, calculated and expressed as GA equivalent, is 28.9 ± 1.4 mg/g of dry plant. Kumar et al. reported a phenolic content 23 mg/g for *Syzygium cumini* leaf extract [72] and Asimuddin et al. reported the value 32.1 mg/g for *Ziziphus mauritiana* leaf extract [73].

The polyphenols consumption is 1.41 ± 0.42 mg, representing ~ 5% of the total amount of phenolic content. This indicates that the concentration of the extract does not influence the reaction rate, instead it influences the size and stability of the formed AgNPs.

#### 3.1.6. Reaction Mechanism between Precursors and Plant Metabolites

Polyphenols, thanks to the p electrons of the multiple hydroxyl groups, establish an internal conjugation with aromatic π electrons, through the loss of electrons and protons, with the formation of oxidation products, i.e., quinones and semiquinones. The potential mechanism involved in the reducing action of the main two flavonoids derivates from *Primula officinalis* plant extract, Quercetin and Kaempferol, on Ag cations is shown in Figure 10. In the case of Quercetin flavonoids, the 1,4–benzopyrone skeleton is conserved, the quinone derivatives forming only in the aromatic 2–phenyl ring, while in the case of Kaempferol an extensive conjugation of the 2–phenyl ring with the pyran heterocycle is involved. A similar reaction mechanism of metallic silver formation by reducing the action of polyphenols was recently proposed by Ahmad et al., 2021 [37].

The type and concentration of derivatives contained in the plant extract further determines the antioxidant activity they exhibit in scavenging short–and long–life free radicals.

#### 3.1.7. In Vitro Determination of the Antioxidant Activity

The CL profile of the extract and AgNPs (Figure 11) showed a strong ability of all samples to scavenge short-life free radicals. When comparing the extract samples to the AgNPs dispersion, a decrease in the antioxidant activity is observed, becoming more intense as the ratio extract:AgNO_3_: is increasing. The percentage of inhibition of the short-life free radicals was in the range 85–96% for the extract samples and 60–94% for the AgNPs dispersions. On the other hand, the TEAC method revealed an increase in the antioxidant activity for the samples containing AgNPs compared to the extract samples, su-ggesting a better ability of scavenging long-life cationic radicals when AgNPs are present (Figure 12). The calculated Trolox equivalent of the pure extract was 39 ± 5 µmol/g of dry plant.

Even though other studies reported an increase in the antioxidant activity via che-miluminescence assay [19,74], it is important to mention that the solutions were diluted 10 times with distilled water prior to chemiluminescence analysis, in order to achieve vi-sible differences between the results, as for all undiluted solutions, the antioxidant activity recorded exceeded 99%. Consequently, this behavior might be due to the stronger ability of the extract to scavenge short-life free radicals compared to that of the AgNPs.

The Trolox equivalent obtained for *Primula officinalis* was higher than those reported by Malinowska (2013) who studied the antioxidant activity of 10 plant extracts used in commercial cosmetics and determined the Trolox equivalent with values situated between 2.28 µmol Trolox/g of extract (for *Centella asiatica L.*) and 20.53 µmol Trolox/g of extract (for *Arnica montana L.*) [75]. In another study, Lizcano and colab. reported the Trolox equivalent for different parts (bark, leaf, stem, fruit) of the plants with medicinal use from Colombian Amazonian and reported values from 1.1 for the fruit of *Crescentia cujete* to 117.4 µmol Trolox/g for the bark of *Brownea rosademonte* [76]. Comparing the ABTS cation radical inhibition activity of AgNPs with the native extract, an increase in the radical sca-venging capacity of nanosilver dispersions was found, which agrees with the previous reports [20,77].

In conclusion, the silver nanoparticles manifest a good scavenger activity for ROS and ABTS cation radicals. This behavior can be explained by the nanosize of AgNPs (demonstrated by DLS, and TEM analysis), the phenolic content of AgNPs dispersions (Figure 10a) and their good physical stability (value of ξ) [19].

### 3.2. Characterization of Textile Samples Treated with AgNPs Dispersions

The effect of nano–silver dispersion (ratio 1:3) on cotton and wool samples was eva-luated in terms of morphological, chromatic, and antimicrobial activity compared to untreated samples.

#### 3.2.1. Morphological Evaluation and Nature Confirmation of the AgNPs Deposited on Textile

SEM characterization was conducted to evaluate the distribution of the AgNPs on the textile fibers (Figure 13) and their nature was confirmed by EDX (Figure 14). Untreated cotton and wool samples were also characterized for comparation. The micrographs revealed that the AgNPs adhered uniformly at the surface of the textile fiber, without deteriorating it.

#### 3.2.2. Chromatic Characterization

After treating the cotton and wool fabrics with AgNPs dispersions, they were chromatically characterized by measuring the L* a* b* parameters, listed in Table 1. Figure 15 shows the color shift from the control (untreated sample) to the treated one. The total color change (ΔE*) was calculated, and the values obtained are 11.29 for cotton and 6.65 for wool. Results indicate an insignificant color influence produced after treating the textile samples with the AgNPs. This aspect is a validation that the treatment meets the conservation criteria of not changing the color of the heritage objects.

The value of the total color difference of cotton sample treated with AgNPs dispersion (11.29) is comparable to that reported by Ibrahim H.M.M. et al. (2016) for cotton treated with AgNPs synthesized using *Alternaria alternata* fungi strain (13.4), when the suspension with 1 mM AgNPs concentration was applied. They have also demonstrated that the color shift is strongly influenced by the concentration of the AgNPs suspension, varying from 13.4 to 41.8 (for 5 mM AgNPs concentration) [78]. The same conclusion was drawn by Kelly et al. and Mahmud and colab. when treating wool samples with AgNPs synthesized using trisodium citrate and sodium alginate, respectively obtaining values higher than 7.06 [27,79].

#### 3.2.3. Antimicrobial Activity of Textile

Textile samples treated with silver nanoparticles were tested to evaluate the antimicrobial activity against three types of bacteria: *Bacillus subtilis, Escherichia coli, Saphylococcus aureus* and one species of fungus *Penicillium hirsutum*. The percentages of bacterial reduction were higher than 99% for all bacteria strains used in the analysis for both cotton and wool samples (Table 2). The qualitative assessment revealed, as well, the formation of a clear zone of inhibition, displayed in Table 3 and compared in Figure 16.

Antimicrobial activity depends on the composition of textile materials and microbial type. The sensitivity of microbes to AgNPs, according to the inhibition zone, was in the following order: *Penicillium hirsutum* > *Bacillus subtilis* > *Escherichia coli* > *Saphylococcus aureus*, for cotton and *Escherichia coli* > *Penicillium hirsutum* > *Saphylococcus aureus* > *Bacillus subtilis* for wool.

It was observed that the sensitivity of fungal strain was higher compared to bacteria. Similar results were reported by Pietrzak and co–workers in their study regarding the antimicrobial activity of silver nanoparticles misting on cotton samples, where the percentage of microorganism’s reduction was 93.42% in the case of *Aspergillius niger* fungal strain, compared to 32.64% for *Bacillus subtilis*, 73.75% for *Escherichia coli*, and 81.98% for *Saphylococcus aureus* bacteria strains [31]. On the other hand, the wool treated samples exhibited a much higher antibacterial activity and a moderately lower antifungal activity. The values of the inhibition zone for wool treated samples were all higher than those reported by Hassan et al., 2 mm for *Staphylococcus aureus*, and 3 mm for *Escherichia coli* [80].

The values of the inhibition zone for cotton are at the same levels as those obtained by Salem et al. in their study about the bactericidal effect of AgNPs produced using leaves of *Mentha longifolia L.* Plant [81].

A greater efficiency of Gram–negative bacteria (*Escherichia coli)* is observed compared to Gram–positive bacteria (*Saphylococcus aureus* and *Bacillus subtilis)* for wool. This beha-vior is in accordance with the results obtained by Barbanta, which demonstrated that at direct contact between silver nanoparticles and bacterial cell resulting deterioration of cell walls and membranes leading to cell death [82]

Consequently, the high percentages of bacterial reduction obtained in all cases (>99%), emphasize the antimicrobial effectiveness of the AgNPs produced in this study as treatment, for textiles preservation against microbial degradation.

## 4. Conclusions

To the best of our knowledge, *Primula officinalis* plant extract was not previously used for AgNPs fabrication with application for preserving textiles. Accordingly, this work provided an attractive method of AgNPs–based treatment fabrication, in full agreement with the concept of biomass valorization for environment-friendly approaches. The synthesis reaction involved aqueous plant extract, whose generous content of phytochemicals played the role of reducing agents. The formation of the green AgNPs was monitored optically, by conducting UV–Vis absorption spectroscopy and optimized by DLS technique and zeta potential measurements. It was demonstrated that the dispersions prepared in various extract: Ag^+^ ratios were physically stable. And the size of the AgNPs was determined by DLS and TEM techniques. The aqueous plant extract exhibited a high ability to scavenge both short-life and long–life free radicals when performing chemiluminescence measurements and TEAC assessment, respectively. In the case of the long-life free radicals, this ability was exceeded by the dispersions containing AgNPs, while chemiluminescence profiles appeared to be slightly decreased when AgNPs were present. The phenolic content of both extract solutions and AgNPs dispersions was determined by Folin-Ciocâlteu method, and a consumption of this was calculated. The AgNPs synthesized with the optimal ratio extract to silver ions were further investigated for their structural (XRD) and morphological (TEM) characteristics. 

According to the chromatic parameters, expressed in the CIE L*a*b* system of colors, an insignificant color influence produced after treatment of the cotton and wool samples with the AgNPs was detected; for instance, the color shift was slightly higher for the cotton compared to wool sample. In addition, SEM characterization of the textile samples ensured that the integrity of the fibers was not affected by the applied treatment and the EDX spectroscopy technique confirmed the presence of AgNPs at their surface. These results represent a validation that the treatment meets the conservation criteria of not chan-ging the color of the heritage objects. The antimicrobial effectiveness of the AgNPs dispersions against bacteria and fungi strains demonstrated notable biocidal properties.

## Figures and Tables

**Figure 1 materials-15-07695-f001:**
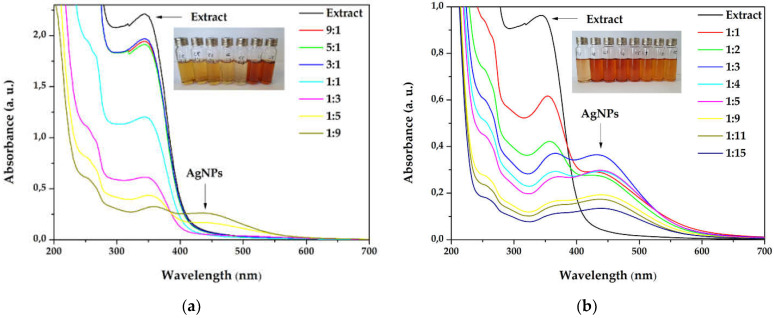
Overlapped ultraviolet–visible (UV–Vis) absorption spectra of silver nanoparticles dispersions at different ratios extract to AgNO_3_ (*v*/*v*) for (**a**) extract concentration 3g/100 mL (**b**) extract concentration 1g/100 mL.

**Figure 2 materials-15-07695-f002:**
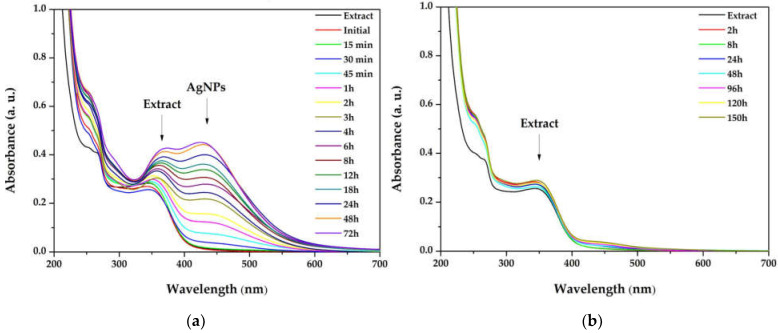
Overlapped ultraviolet–visible (UV–Vis) absorption spectra of silver nanoparticles dispersions at different times of the synthesis, performed in (**a**) simulated sunlight and (**b)** darkness.

**Figure 3 materials-15-07695-f003:**
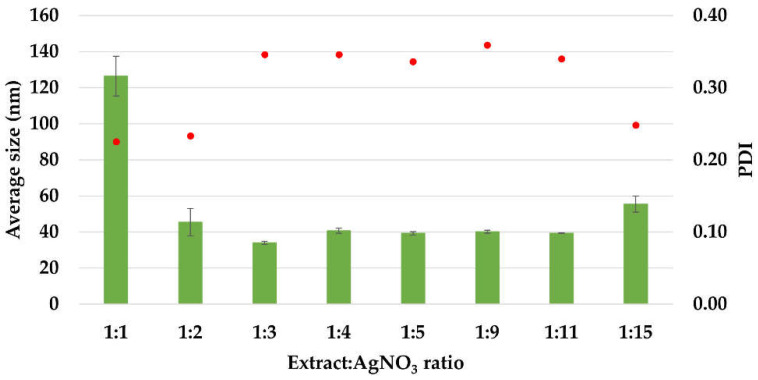
The average size (Z_ave_, nm) and polydispersity index (PDI) of AgNPs, estimated by Dynamic Light Scattering (DLS) measurements.

**Figure 4 materials-15-07695-f004:**
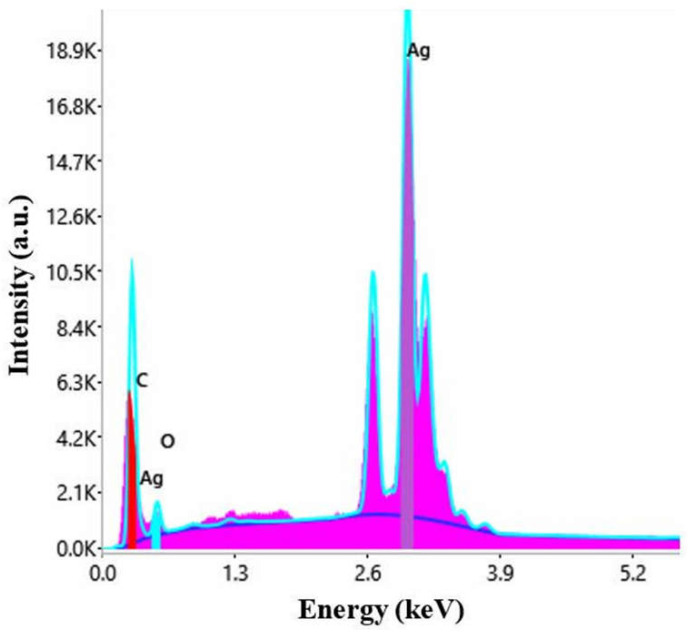
EDX spectrum of AgNPs synthesized with *Primula officinalis* extract.

**Figure 5 materials-15-07695-f005:**
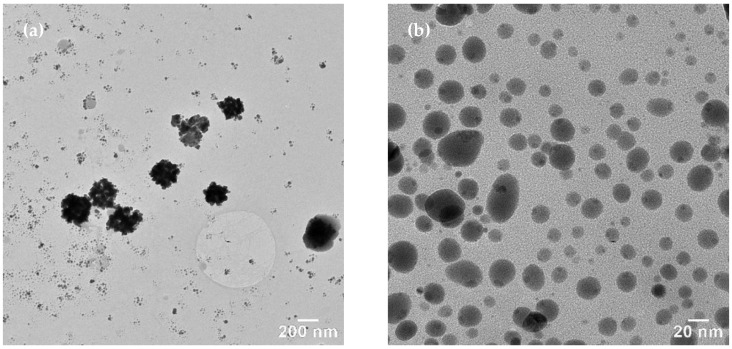
TEM images of AgNPs synthesized with *Primula officinalis* extract 1:3 (**a**), clustered AgNPs synthesized with *Primula officinalis* extract 1:3 (**b**), the particle size distribution as obtained on cvasi–spherical Ag NPs (**c**).

**Figure 6 materials-15-07695-f006:**
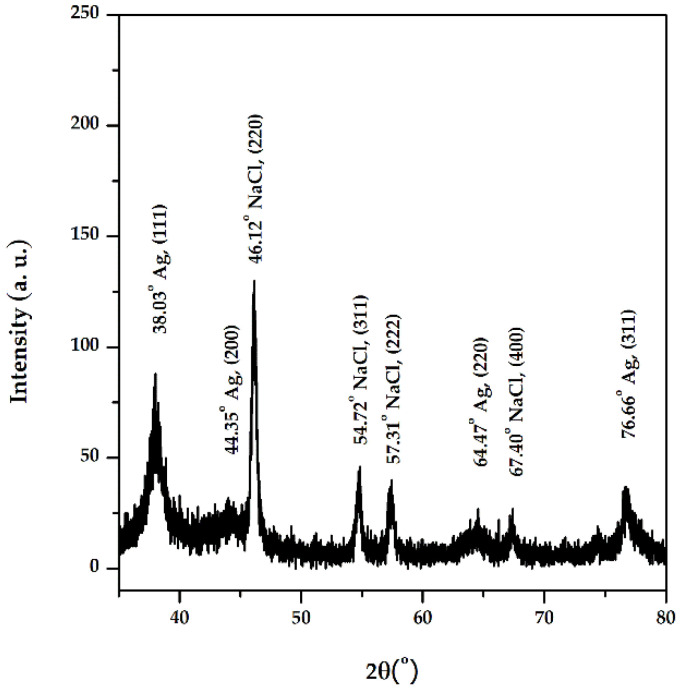
XRD diffractogram of AgNPs synthesized with *Primula officinalis* extract.

**Figure 7 materials-15-07695-f007:**
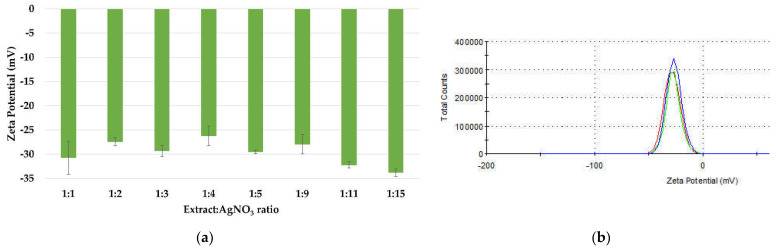
Zeta potential measurements of the AgNPs dispersions (**a**) and the distribution of the zeta potential (**b**).

**Figure 8 materials-15-07695-f008:**
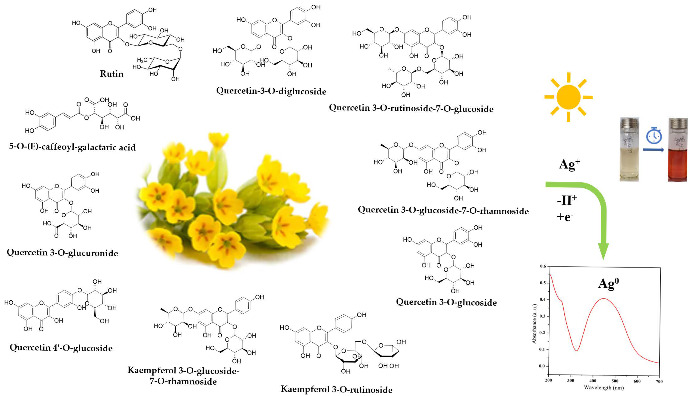
Schematic representation of AgNPs green synthesis using *Primula officinalis* plant extract.

**Figure 9 materials-15-07695-f009:**
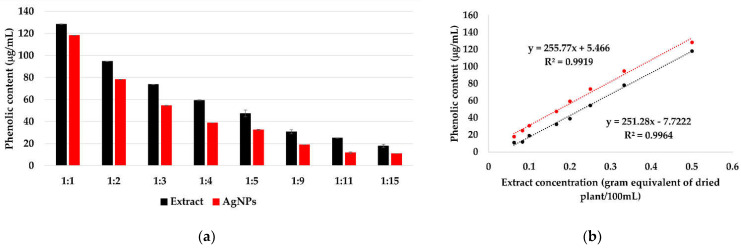
Total polyphenols in extract solutions and AgNPs dispersions, determined by Folin-Ciocâlteu method (**a**). The dependency of the phenolic content on the extract concentration in extract solutions and AgNPs dispersions (**b**).

**Figure 10 materials-15-07695-f010:**
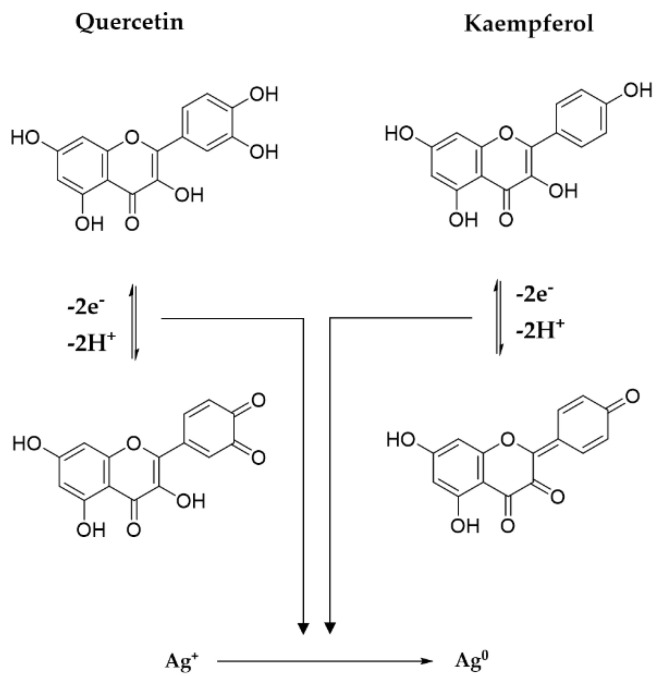
Mechanism of metallic silver formation.

**Figure 11 materials-15-07695-f011:**
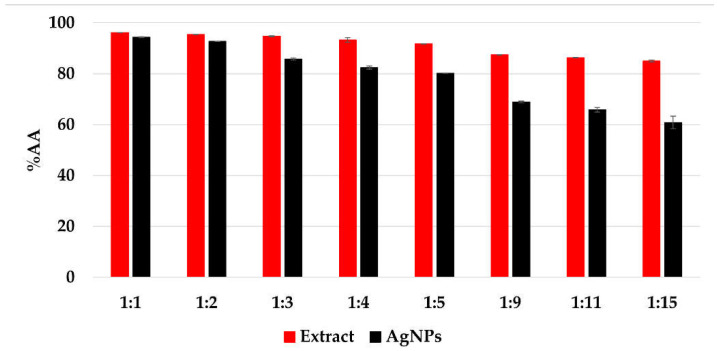
The antioxidant activity (%*AA*) of the samples evaluated by chemiluminescence method.

**Figure 12 materials-15-07695-f012:**
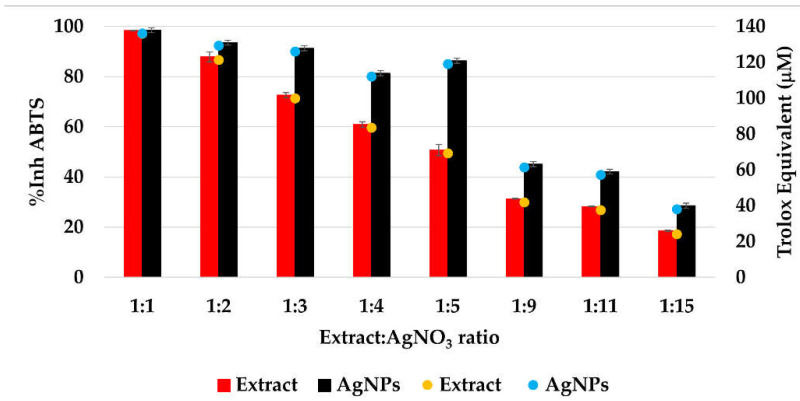
The antioxidant activity (%*AA*) of the samples evaluated by TEAC method.

**Figure 13 materials-15-07695-f013:**
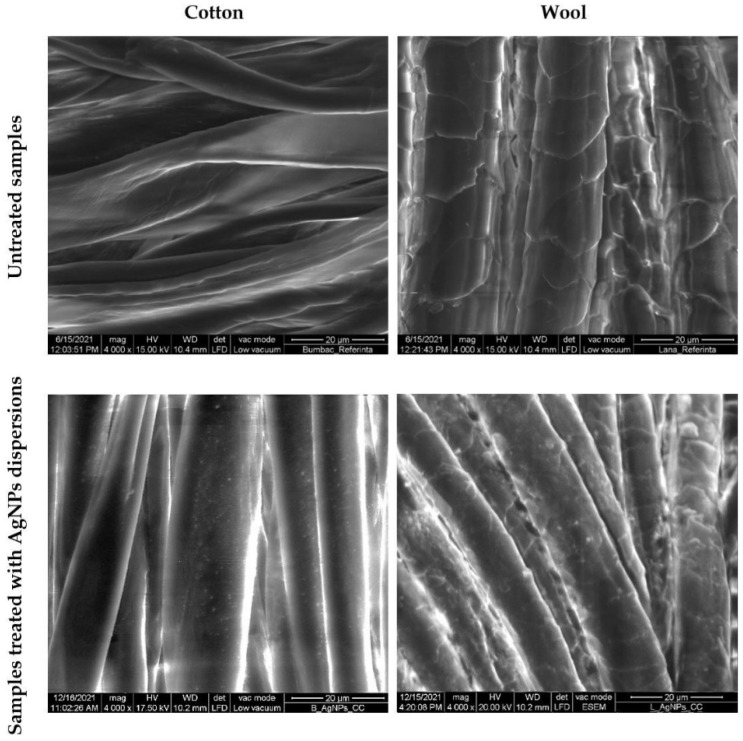
SEM images of cotton and wool samples treated with AgNPs dispersions.

**Figure 14 materials-15-07695-f014:**
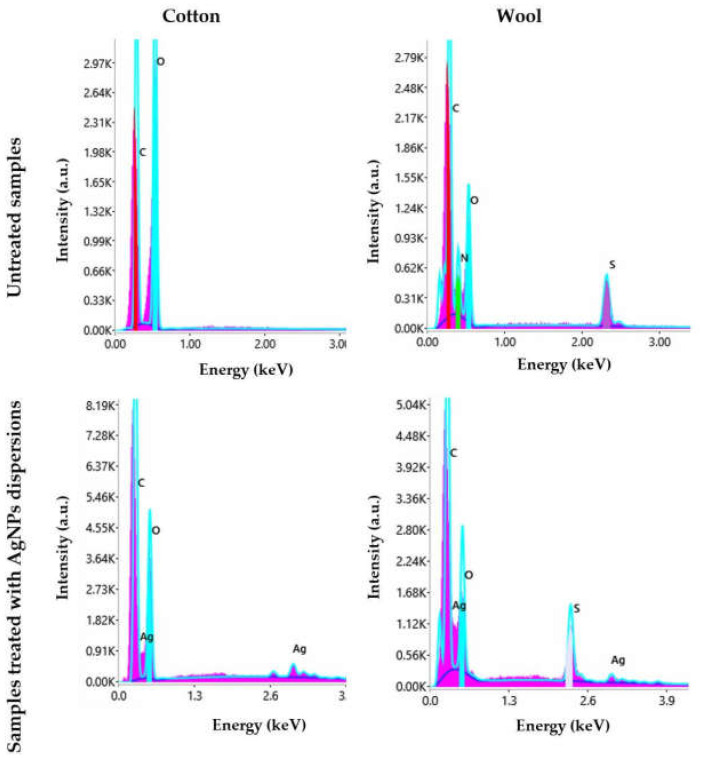
EDX spectra of cotton and wool samples treated with AgNPs dispersions.

**Figure 15 materials-15-07695-f015:**
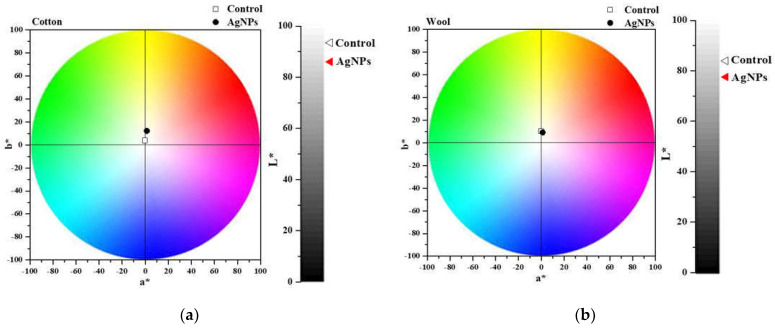
Chromatic diagrams of cotton(**a**) and wool (**b**) samples treated with AgNPs dispersions.

**Figure 16 materials-15-07695-f016:**
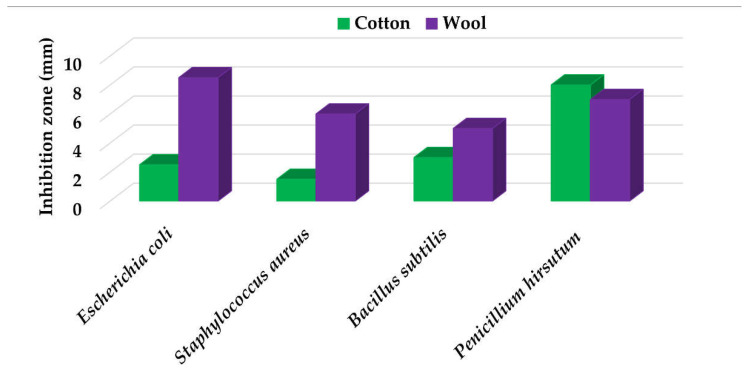
Inhibition zone of Petri dishes inoculated with microbial strains and incubated with AgNPs treated textile samples.

**Table 1 materials-15-07695-t001:** Chromatic parameters of cotton and wool samples treated with AgNPs dispersions.

Sample		L*	a*	b*	∆L*	∆a*	∆b*	∆E*
Untreated Cotton		93.48	−0.27	3.83	−	−	−	−
Untreated Wool		83.96	−0.19	10.17	−	−	−	−
AgNPs	Cotton	86.01	1.35	12.14	−7.47	1.62	8.1	11.29
	Wool	77.54	1.16	9.12	−6.42	1.34	−1.05	6.65

**Table 2 materials-15-07695-t002:** Colony–forming units of the bacteria strains and bactericidal ratios for textile samples treated with AgNPs dispersions.

Bacterial Strain	Textile Sample	CFUs/mL on Control Samples	CFUs/mL on Samples Treated with AgNPs Dispersions	Bactericidal Ratio (%)
*Escherichia coli*	Cotton	2.1 × 10^4^	11	99.94
	Wool	2.8 × 10^4^	7	99.97
*Saphylococcus aureus*	Cotton	4.5 × 10^4^	9	99.98
	Wool	5.5 × 10^4^	11	99.98
*Bacillus subtilis*	Cotton	2.9 × 10^4^	4	99.99
	Wool	3.7 × 10^4^	6	99.98

**Table 3 materials-15-07695-t003:** Images of Petri dishes inoculated with microbial strains and incubated with AgNPs treated textile samples.

Textile Sample	Cotton	Wool
Microbial strain		
*Escherichia coli*	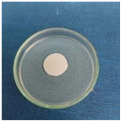	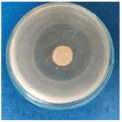
*Saphylococcus aureus*	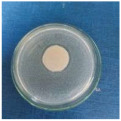	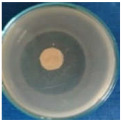
*Bacillus subtilis*	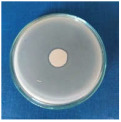	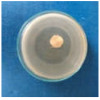
*Penicillium hirsutum*	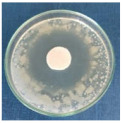	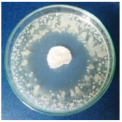

## Data Availability

The data were included in the text.

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
