# Peer review of "Design of Green Silver Nanoparticles Based on Primula Officinalis Extract for Textile Preservation"

_materials, 2022, doi:10.3390/ma15217695_

Round 1
Reviewer 1 Report
This manuscript entitled "Design of green silver nanoparticles based on Primula officinalis extract for textile preservation" investigates the characterization of the synthesized material and its potential application on antimicrobial treatment for textile preservation. This paper presents a good idea that could be used in different applications in water and wastewater treatment, drug delivery systems and other potential application. Below are few comments author can address to make this paper better.
1. Abstract should be more lucrative. Provide more quantitative results/findings. Remove repetition of words such as “and and” from the abstract.
2. The English grammar is not up to mark. Manuscript should be checked based on grammar.
3. The novelty of the work must be presented clearly in both abstract and introduction.
4. You mentioned good details about the preparation and characterization of the chemicals (in the Materials and methods section), but section 2.3.3. Phenolic content assessment is without referenced. You must mention the standards or the reference of this.
5. Characterization of Primula officinalis extract should be done in order to confirm its constituents. The actual constituent of the extract can be use in the reaction mechanism.
6. Better to have separate section of mechanism of green synthesis, reaction between precursors and plant metabolites.
7. It would be better if you write conclusion in one or hardly in two paragraphs. No need to write the values in conclusion, better to write the opinion or conclusion or gist of the results coming out from the experimentation. Avoid writing values as you have already discussed this thing in “results and discussion”. Sate the main findings only.
Author Response
Dear Reviewer,
We sincerely thank you for the helpful comments and suggestions that contributed to further improve the manuscript “Design of green silver nanoparticles based on Primula officinalis extract for textile preservation”. The manuscript has been carefully reviewed based on these comments.
All comments provided by the reviewer have been addressed and presented in the revised manuscript with a yellow background.
REVIEWER 1
- Abstract should be more lucrative. Provide more quantitative results/findings.
Author’s answer: More quantitative results/findings were inserted in the Abstract. This section was partially rephrased to avoid exceeding the limit number of words.
- The English grammar is not up to mark. Manuscript should be checked based on grammar.
Author’s answer: English grammar was rechecked.
- The novelty of the work must be presented clearly in both abstract and introduction.
Author’s answer: The novelty of the research was presented in both sections.
- You mentioned good details about the preparation and characterization of the chemicals (in the Materials and methods section), but section 2.3.3. Phenolic content assessment is without referenced. You must mention the standards or the reference of this.
Author’s answer: The standard of phenolic content assessment was mentioned and cited.
- Characterization of Primula officinalis extract should be done in order to confirm its constituents. The actual constituent of the extract can be use in the reaction mechanism.
Author’s answer:
The Primula officinalis extract characterization was carried out in our research from the point of view of the total content of polyphenols because the behaviour of these active phytochemicals (eg: the quantitative amount of polyphenols needed to reduce the silver ion) was correlated to the extract reducing capacity. Thus, an amount of 28.9 ± 1.4 mg/g of dry Primula officinalis plant was determined as gallic acid equivalent. In a related study (Tarapatskyy 2021), the main polyphenolic compounds from Primula veris (as known as Primula officinalis) are consisting in 18 Quercetin- and Kaempferol-glycosidic derivatives. These informations have been introduced in the revised manuscript.
- Better to have separate section of mechanism of green synthesis, reaction between precursors and plant metabolites.
Author’s answer: A separate section of mechanism of green synthesis, reaction between precursors and plant metabolites was included.
- It would be better if you write conclusion in one or hardly in two paragraphs. No need to write the values in conclusion, better to write the opinion or conclusion or gist of the results coming out from the experimentation. Avoid writing values as you have already discussed this thing in “results and discussion”. Sate the main findings only.
Author’s answer: The “Conclusion” section has been shortened and the main outcomes of the
work have been emphasized.
In summary, all the comments and suggestions provided by the Reviewers were implemented. These are highlighted in the document with yellow background, and we hope that they have made considerable improvements to the manuscript.
Yours sincerely,
Prof. dr. Nicoleta Badea

Reviewer 2 Report
1. Please recheck the optical data evaluation
2. two peaks in XRD are not marked
3. why does the TEM of your sample show such diverse particle sizes? Is it possible to control the particle size to make it monodisperse using this technique
4. Very good works on Silver nanoparticles they are very highly cited please refer them and improve your introduction using them
a. Rapid synthesis of silver nanoparticles using dried medicinal plant of basil
N Ahmad, S Sharma, MK Alam, VN Singh, SF Shamsi, BR Mehta, A Fatma
Colloids and Surfaces B: Biointerfaces 81 (1), 81-86, 2010
b. Biosynthesis of silver nanoparticles from Desmodium triflorum: a novel approach towards weed utilization
N Ahmad, S Sharma, VN Singh, SF Shamsi, A Fatma, BR Mehta
Biotechnology research international 2011, 454090
c. Ocimum mediated biosynthesis of silver nanoparticles
N Ahmad, MK Alam, VN Singh, SF Shamsi, S Sharma
2009 Fifth International Conference on MEMS NANO, and Smart Systems, 80-84
Author Response
Dear Reviewer,
We sincerely thank you for the helpful comments and suggestions that contributed to further improve the manuscript “Design of green silver nanoparticles based on Primula officinalis extract for textile preservation”. The manuscript has been carefully reviewed based on these comments.
All comments provided by the reviewer have been addressed and presented in the revised manuscript with a yellow background.
- Please recheck the optical data evaluation
Author’s answer: After rechecking the optical data evaluation, the UV-Vis spectra were completed by marking the absorption bands corresponding to materials.
- Two peaks in XRD are not marked
Author’s answer: The diffraction lines in the XRD figure were assigned accordingly.
- Why does the TEM of your sample show such diverse particle sizes? Is it possible to control the particle size to make it monodisperse using this technique
Author’s answer: The AgNPs obtained with 1:3 Primula officinalis extract 1:3 shows mainly a monodisperse, quasi-spherical system of Ag NPs. However, the presence of much larger Ag clusters was randomly observed”.
- Very good works on silver nanoparticles they are very highly cited please refer them and improve your introduction using them
Author’s answer: The recommended articles contain valuable data, relevant to the present study and they were cited in the “Introduction” section.
In summary, all the comments and suggestions provided by the Reviewers were implemented. These are highlighted in the document with yellow background, and we hope that they have made considerable improvements to the manuscript.
Yours sincerely,
Prof. dr. Nicoleta Badea

Reviewer 3 Report
Comments to the Authors
1. In this manuscript authors prepared silver nanoparticles (AgNPs) using Primula officinalis plant extract and the phenolic consumption was established using Folin-Ciocâlteu method and the antioxidant activity was evaluated using chemiluminescence and Trolox Equivalent Antioxidant Capacity methods/TEAC methods. This research has value for the researchers in the related areas. However, the paper needs improvement before acceptance for publication. My detailed comments are as follow:
1. In the introduction section authors should discuss the basic properties of silver nanoparticles. For this authors should use the following articles as reference:
a. doi.org/10.1002/slct.201900470
b. doi.org/10.1016/j.jece.2020.104596
2. Authors should provide a histogram of size of AgNPs obtained from TEM images.
3. In the Figure 12, authors should mark the AgNPs on the cotton and wool samples treated with AgNPs dispersions as it is not visible.
4. In the UV-visible spectra authors should marked the peaks corresponding to materials.
5. There are lots of typos errors like “ chemiluminescence and and Troloxin” the abstract and also other portions.
Author Response
Dear Reviewer,
We sincerely thank you for the helpful comments and suggestions that contributed to further improve the manuscript “Design of green silver nanoparticles based on Primula officinalis extract for textile preservation”. The manuscript has been carefully reviewed based on these comments.
All comments provided by the reviewer have been addressed and presented in the revised manuscript with a yellow background.
REVIEWER 3
- In the introduction section authors should discuss the basic properties of silver nanoparticles. For these authors should use the following articles as reference:
- doi.org/10.1002/slct.201900470
- doi.org/10.1016/j.jece.2020.104596
Author’s answer: We thank you for your recommendations. The articles were used to enrich the references section and to add extensive value to our paper.
- Authors should provide a histogram of size of AgNPs obtained from TEM images.
Author’s answer: A size histogram was inserted.
- In the Figure 12, authors should mark the AgNPs on the cotton and wool samples treated with AgNPs dispersions as it is not visible.
Author’s answer: SEM images were processed to enhance the white “dots” that represent the AgNPs.
- In the UV-visible spectra authors should marked the peaks corresponding to materials.
Author’s answer: The absorption bands in the UV-visible spectra were assigned corresponding to materials.
- There are lots of typo’s errors like “chemiluminescence and and Troloxin” the abstract and also other portions.
Author’s answer: The paper was carefully rechecked for typos, and they corrected.
In summary, all the comments and suggestions provided by the Reviewers were implemented. These are highlighted in the document with yellow background, and we hope that they have made considerable improvements to the manuscript.
Yours sincerely,
Prof. dr. Nicoleta Badea

Round 2
Reviewer 1 Report
The authors have made sufficient modifications, and I suggest that this paper be accepted without further modification.
Reviewer 2 Report
Accept